# Healthcare Resource Use and Costs of Allogeneic Hematopoietic Stem Cell Transplantation Complications: A Scoping Review

**DOI:** 10.3390/curroncol32050283

**Published:** 2025-05-16

**Authors:** Nancy V. Kim, Gemma McErlean, Serena Yu, Ian Kerridge, Matthew Greenwood, Richard De Abreu Lourenco

**Affiliations:** 1Centre for Health Economics Research and Evaluation, University of Technology Sydney, Chippendale 2008, Australia; 2School of Nursing, University of Wollongong, Wollongong 2522, Australia; 3Ingham Institute for Allied Health Research, Liverpool 2170, Australia; 4St George Hospital, South Eastern Local Health District, Kogarah 2217, Australia; 5Department of Hematology, Royal North Shore Hospital, St Leonards 2065, Australia; 6Northern Clinical School, Faculty of Medicine and Health, University of Sydney, Camperdown 2006, Australia; 7Northern Blood Research Centre, Kolling Institute, St Leonards 2064, Australia

**Keywords:** Allogeneic hematopoietic stem cell transplant, costs, health care resource utilization

## Abstract

Allogeneic hematopoietic stem cell transplant (allo-HSCT) is an expensive and resource intensive procedure. This study aims to review the literature pertaining to healthcare resource utilization (HRU) and costs associated with allo-HSCT complications. The review followed the Joanna Briggs Institute methodology for scoping reviews. The PubMed, EMBASE, and Health Business Elite were searched in addition to the grey literature. Eligibility criteria included studies that reported HRU and/or costs associated with adult (≥18 years) allo-HSCT. Studies were categorized according to complications of allo-HSCT including graft-versus-host disease (acute and chronic GVHD) and infections (fungal, cytomegalovirus, virus-associated hemorrhagic cystitis, and acute respiratory tract infection). Commonly reported HRU and cost measures were extracted, including those associated with the direct management of allo-HSCT complications and intensive care unit (ICU) admissions. Reported costs were standardized to 2022 United States Dollars. Patients who experienced GVHD or infection post-transplant had an overall greater HRU including higher rates of hospitalization, hospital readmission, ICU admission, and longer length of stay compared to those patients who did not. Patients with severe or refractory GVHD and/or infection following allo-HSCT required greater healthcare intervention. This scoping review synthesizes the current literature on HRU and costs associated with post allo-HSCT complications. Patients who experienced post allo-HSCT complications had higher HRU and incurred higher costs overall, noting the variability across studies in their clinical context, reporting of HRU, and cost measures.

## 1. Introduction

Allogeneic hematopoietic stem cell transplant (allo-HSCT) provides effective treatment for many malignant and non-malignant diseases. The number of allo-HSCTs performed each year continues to increase worldwide [1]. In 2021, 8436 allo-HSCTs were performed in the US [2] and 19,806 in Europe [3], an approximate 2.0% and 5.3% increase respectively, compared to the previous year.

Over the past three decades, there have been significant improvements in outcomes post-allo-HSCT [4,5]. Despite these improvements, allo-HSCT is still associated with significant morbidity and mortality, particularly related to graft versus-host-disease (GVHD) and infection [6]. The clinical impact of GVHD includes higher hospitalization rates, increased risk of severe infection and increased mortality [7]. Opportunistic infections following allo-HSCT, such as cytomegalovirus (CMV) and fungal infections, are also common and potentially life-threatening [8]. CMV infection is associated with an increased overall and non-relapse mortality [9], whilst invasive fungal infection is associated with a high attributable mortality rate [10]. 

Allo-HSCT itself is a resource-intensive and highly expensive therapy [11]. Post-transplant complications further increase health resource utilization (HRU) and costs [12,13]. But while it is widely recognized that post-transplant complications, particularly infection and GVHD, are major determinants of mortality and morbidity post-allo HSCT [14,15] there is limited evidence regarding their impact on HRU and costs [16]. Understanding the resources needed to prevent and manage complications is critically important for clinicians, policymakers, and governments, as it may inform decisions about the efficient allocation of health care resources for allo-HSCT [17,18]. 

The aim of this study is to review the literature for HRU and costs associated with complications that contribute most to excess mortality and morbidity post-allo-HSCT—infection and GVHD. The review expands on an earlier scoping review that examined overall HRU and costs associated with allo-HSCT [11].

## 2. Methods

This literature review followed the Joanna Briggs Institute (JBI) methodology for scoping reviews [19] and aligns with the Preferred Reporting Items for Systematic Review and Meta-Analyses extension for Scoping Reviews (PRISMA-ScR) checklist [20]. As the aim of the literature review was to provide an overview of the evidence, a scoping review was considered appropriate. In accordance with the standard methods of a scoping review, a critical appraisal of the studies included was not performed [19,21]. A prospective Study Protocol was published on Open Science Framework (https://osf.io/5tdsw/).

Eligibility criteria included studies of primary research written in English, with full-text availability. Studies that reported costs solely linked to HSCT registries and the pre-transplant phase (e.g., donor screening) without focusing on a specific complication related to allo-HSCT did not fulfill the criteria. Specific population, concept, and context (PCC) [19] eligibility included the following: 

*Population*—adult (≥18 years) recipients of allo-HSCT (age as reported by study eligibility criteria, or baseline age range or standard deviation).

*Concept*—studies that reported HRU and costs from the health system perspective were included. Studies only presenting indirect or nonmedical costs, e.g., productivity and out-of-pocket costs, were excluded. 

*Context*—cost studies from high-income countries as classified by the World Bank Country and Lending Groups [22] were included to ensure comparability in terms of access and affordability [23]. 

The PubMed, EMBASE, and Health Business Elite databases were searched from inception until November 2022 using key terms related to ‘allogeneic hematopoietic stem cell transplantation’ and ‘health care resource utilization’. The grey literature was searched using the relevant subsections of the Health Technology Assessment (HTA) Agencies and Health Economics sections from the Canadian Agency for Drugs and Technologies in Health (CADTH) “Grey Matters: a practical tool for searching health-related grey literature” checklist [24]. The MEDLINE search strategy is presented in Appendix A.

Data screening and extraction were conducted by the principal reviewer (NVK), with independent screening of 20% of the sample by a second reviewer (GM) at each of two stages of screening (abstract and full text). Interrater reliability (IRR) was calculated at each stage, with an acceptability threshold of greater than 80% agreement, measured by the prevalence-adjusted bias-adjusted kappa (PABAK) [25]. Data extraction was performed by NVK using a modified version of the JBI template instrument for extraction [11] (Appendix A). A 20% sample of the extracted data was audited by GM.

### Data Analysis

Included studies were categorized by those that reported on allo-HSCT associated complications of GVHD and infections (fungal, CMV, virus-associated hemorrhagic cystitis (V-HC) and acute respiratory tract infection (ARTI)). Studies were stratified according to the type of complication and management. For GVHD, studies were stratified according to whether they reported prophylaxis or treatment of acute (aGVHD) and/or chronic GVHD (cGVHD), while studies of allo-HSCT associated infection were stratified according to whether they reported CMV or fungal infection, and where mentioned, if they focused on prophylaxis, pre-emptive therapy (Pre-ET) or treatment. 

HRU and cost measures as reported within the included studies were extracted for this review according to the allo-HSCT complication of interest. As per the earlier scoping review [11], extracted HRU and cost measures were length of stay (LOS), outpatient visits, hospital admission and readmissions. Common metrics used to express transplant-associated costs and HRU included total cost, total hospitalization, intensive care unit (ICU) admission, hospital readmission, post discharge, drug and laboratory costs. 

Where possible, HRU and cost measures reported on a comparable basis were combined and presented as a mean and/or range across the included studies. Comparability across the studies was based on the clinical context (e.g., prophylaxis, Pre-ET, and treatment), the measure of HRU or cost reported (based on the measures above), and the time point for reporting (e.g., duration post-allo-HSCT). Where output measures could not be synthesized across studies, HRU and cost measures were presented graphically without aggregation of results. Results are presented based on comparable time points across studies (e.g., months’ post-transplant, 100-days, 1 year, study period). Finally, a qualitative summary synthesizing the HRU and cost results from the included studies is reported.

Where studies reported both the mean and median, the mean values were used within this paper. Cost results presented by studies were standardized to 2022 United States Dollars (USD) with monetary values expressed in other currencies converted to USD based on the exchange rate of the study period (Board of Governors of the Federal Reserve System [26]) and inflated to 2022 prices (United States Bureau of Labor Statistics Consumer Price Index [27]). Where a study did not specify the time value or currency, the country of the primary author and the year of study publication were used as reference.

## 3. Results

### 3.1. Literature Search

From a total of 500 full-text reviews 38 studies focused on GVHD or infection complications associated with allo-HSCT (Figure 1).

The PABAK met the acceptability threshold for title and abstract screening of the database search but not for the full-text screen. The study team accepted the level of agreement (71.4% compared to a threshold of 80%) as the discrepancy was considered amplified by the small sample size (4 discrepancies of 28 studies reviewed), and consensus was achieved amongst the two reviewers. The PABAK was not calculated for the grey literature search; however, there was agreement amongst the two reviewers regarding the documents included.

#### Characteristics of Included Studies

Most studies were single-center (14, 36.8%) or cohort-based (14, 36.8%) with few multicenter studies (3, 7.9%). The remaining were economic evaluations (5, 13.2%) and Government Health Technology Assessment Reports (2, 5.3%). The majority originated in North America (37.3%), followed by Europe (27.5%) (Appendix A). Over half (60.5%) were published in the last 5 years (i.e., since 2020).

Allo-HSCT complications reported by the studies included GVHD (11, 28.9%), CMV (19, 50%), fungal infection (5, 9.8%), V-HC (2, 5.3%) and ARTI (1, 2.6%). HRU and cost results from seven (18.4%) studies were based on the partial population from study arms (4, 10.5%) or subgroups (3, 7.9%). The majority (27, 72.2%) presented HRU and/or cost measures at time points comparable to other studies (Table 1); details of the HRU and cost measures reported by the studies are in Appendix A, including those that could not be compared [28,29,30,31,32,33,34,35,36,37,38].

### 3.2. HRU and Costs

#### 3.2.1. GVHD

Of the 11 studies (28.9%) reporting on GVHD, nine (23.7%) reported both HRU and costs, and two (5.3%) reported HRU measures only.

##### Acute GVHD

Four studies (10.5%) [39,40,41,42] presented HRU associated with aGVHD, with three also presenting associated costs (7.9%) [39,40,42].

The duration of initial hospitalization was longer for aGVHD compared to non-GVHD [39,40], with an even longer duration for patients with steroid-refractory or high-risk (SR/HR) aGVHD [40]. Subgroup analyses indicated shorter hospitalization in patients receiving one line compared to ≥2 lines of aGVHD therapy [41]. At 100 days, aGVHD was associated with higher readmission and ICU admission rates compared to non-aGVHD [40,42], with fewer hospitalizations and ICU days among those receiving one prior line of aGVHD therapy compared to ≥2 lines [41] (Figure 2). aGVHD was also associated with a longer total LOS compared to non-aGVHD [42], and outpatient visits were more frequent for those who received one prior line of therapy compared to ≥2 lines [41].

The costs of initial hospitalization were approximately 60–70% higher for aGVHD, compared to non-aGVHD [39,40], with more significant costs associated with SR/HR aGVHD [40]. At 100 days, costs associated with aGVHD were generally higher compared to non-GVHD, including aGVHD-attributed costs, inpatient and outpatient costs [42]. Total cost of readmission was also higher for aGVHD compared to non-GVHD, with higher costs for SR/HR aGVHD [40]. Overall, total costs were higher for patients with aGVHD compared to non-GVHD at 100 days and within the first year post allo-HSCT [42].

##### Chronic GVHD

Two studies (5.3%) [29,43] reported HRU, two (5.3%) [29,44] reported costs associated with cGVHD, and one study both [29]. All three studies defined cGVHD onset at different time points: greater than 100-, 180- and 182-days post allo-HSCT.

The proportion and number of hospitalizations, along with the LOS per admission, at 3 years were higher for cGHVD compared to non-cGVHD, with greater frequency and longer admissions for more severe cGVHD compared to moderate and mild cGVHD [43].

Total cost associated with cGVHD based on the cumulative cost per patient for the first 3 years of follow-up was highest for moderate/severe cGVHD when compared with mild and non-cGVHD, respectively [44].

##### GVHD Treatment

Three studies [30,45,46] (7.9%) presented HRU and costs associated with the treatment of GVHD. Two studies [45,46] looked at HRU and costs associated with Extracorporeal Photopheresis (ECP) for GVHD treatment. At 1 year post steroid-refractory cGVHD, a lower proportion of the ECP group had hospitalizations and shorter LOS, compared to non-ECP recipients, but a higher number of outpatient day-hospital visits and external consultations, resulting in lower inpatient and total costs overall [46]. For complications associated with systemic steroid use for GVHD, the presence of both aGVHD and cGVHD was associated with higher costs compared to either alone [30].

#### 3.2.2. CMV

Among the studies analyzed, those detailing HRU and costs related to CMV management in allo-HSCT were the most common, comprising 19 studies (50%). Of these, ten studies (26.3%) provided information on both HRU and costs, six (15.8%) focused solely on HRU, and three (7.9%) on costs.

##### CMV Prophylaxis

Five studies [47,48,49,50,51] reported the overall costs associated with letermovir prophylaxis in CMV seropositive patients compared with no prophylaxis or standard of care (Figure 3). The average of the total reported costs across the studies was greater for letermovir prophylaxis (USD 29,995) compared to no prophylaxis or standard of care (USD 16,299).

Two [50,51] of the five studies, applied similar model designs (cost-effectiveness analysis based on decision analytical modelling) but based on different regions (United States and Hong Kong). These studies also reported additional cost comparisons between letermovir prophylaxis and no prophylaxis or standard of care, indicating letermovir prophylaxis was associated with lower costs for CMV-related re-hospitalization, CMV Pre-ET and CMV disease (CMVd), but higher costs associated with CMV prophylaxis (compared to zero cost with no prophylaxis or standard of care).

##### CMV Infection or Disease 

Thirteen studies (34.2%) reported HRU and costs associated with CMV infection (CMVi) or CMVd. Five studies (13.2%) focused on CMVi, one (2.6%) on CMVd, and seven (18.4%) on a combination of CMVi/CMVd. Definitions for the CMV group varied across the studies, including “patients with clinically significant CMV”, “CMV infection”, “CMV viremia”, “patients receiving Pre-ET”, and “CMV episodes” (commonly defined as patients with CMVi/CMVd). Criteria for initiating Pre-ET differed among the studies. Most did not specify prophylaxis or noted only some patients received prophylactic therapy.

CMV was associated with longer initial hospitalization compared to no CMV (pre-ET vs. no pre-ET; CMVi vs. without CMVi) [52,53]. This finding was consistent when examining the length of initial stay based on 1 and ≥2 CMV-associated readmissions [54], which were higher than those with no readmissions. LOS was reported in ten studies [38,52,54,56,57,58,59,60,61,62] (Figure 4i) over various timeframes. Most studies compared the CMV population to those without. Overall, a longer LOS was observed with CMV than the respective comparator, though the magnitude varied.

Five studies reported the proportion of patients with readmissions [52,56,57,61,62] and six focused on the number of hospitalizations [38,56,59,60,61,62], while three [56,61,62] reported both. In all the studies, the CMV groups showed higher rates and more readmissions compared to their respective comparators (Figure 4ii). The duration of readmission was longer for CMVi at 180 days (Pre-ET vs. no Pre-ET, respectively) [52], and at 1 year (CMVi during and after allo-HSCT hospitalization vs. without CMVi) [53].

At 1 year, a higher proportion of patients with CMV required ICU admission compared to those without [56]. When stratified by the number of CMVi, more patients with 1 CMVi had ICU admissions compared to those with no CMVi or ≥2 CMVi [62]. The length of ICU admission during the allo-HSCT hospitalization was longer for patients with multiple CMV readmissions compared to those with no CMV readmissions [54]. The proportion of patients with outpatient visits at 180 days was higher for those with 1 and ≥2 CMVi or CMVd [57], and at 1 year for CMVi/CMVd [61] compared to no CMV. The number of outpatient visits at 1 year was also more frequent for CMVi/CMVd [61].

While, seven studies [32,38,54,56,57,61,62] reported aspects of CMV-directed therapy, three were difficult to interpret [54,57,61], as it was unclear whether the antiviral therapies examined were used for prophylaxis or treatment of CMV. Antiviral medications most frequently administered for treatment of CMVi or CMVd were valganciclovir, ganciclovir and foscarnet, although the choice of antivirals varied across studies [32,56,62] and number of CMV episodes [56].

Total cost of allo-HSCT for the CMV population was consistently higher than the respective comparators at various timeframes [53,55,57,58,60,61] (Figure 4iii). When stratified further, total costs at 1 year were slightly higher for CMV seronegative than seropositive recipients, and when the first CMVi occurrence was after 100 days post allo-HSCT compared to within the first 100 days [58].

Overall, inpatient admission costs were higher for the CMVi or CMVi/CMVd population relative to their respective comparators (Figure 4iv). Readmission costs were higher for the CMV group at 180 days (Pre-ET vs. no Pre-ET) [51], and 1 year (CMVi during or after allo-HSCT hospitalization vs. readmission without CMVi [52]. Costs for the CMV group were higher than the no CMV group, in terms of inpatient cost for HSCT admission (Pre-ET vs. no Pre-ET [51], and outpatient costs at 1-year (CMVi/CMVd vs. no CMV) [61]. The cost of antiviral therapy per patient, at 100 days was highest for CMVd, followed by CMVi, with no CMV incurring the lowest costs [54]. 

#### 3.2.3. Fungal Infection

Five studies [33,34,63,64,65] (13.2%) presented HRU, and costs associated with the management of fungal infections. Three studies (7.9%) reported both HRU and costs and two studies (5.3%) reported costs only.

The use of dual antifungal prophylaxis (posaconazole plus micafungin) compared to single-agent micafungin during allo-HSCT hospitalization, resulted in slightly longer initial hospitalization and higher costs [33]. The duration of prophylaxis was shorter for dual therapy compared to single-agent micafungin with similar costs of diagnostic approaches and antifungal therapy [33].

Direct comparison of prophylactic posaconazole and itraconazole at 100 days resulted in higher total cost for posaconazole compared to itraconazole, which included higher antifungal drug costs but slightly lower inpatient costs [64]. Various antifungal management strategies, including posaconazole prophylaxis, empirical voriconazole treatment after fluconazole prophylaxis, and Pre-ET with voriconazole post-fluconazole prophylaxis (based on regular galactomannan testing), resulted in similar costs [65].

#### 3.2.4. Other Infections

LOS and rate of readmission were higher for patients with V-HC compared to those without, resulting in higher total costs at 1 year [35]. Patients with ARTI undergoing cord or allo-HSCT showed a longer LOS, resulting in higher post-HSCT hospitalization costs compared with non-ARTI [37].

## 4. Discussion

This scoping review provides an overview of HRU and costs associated with the most prevalent and morbid complications of allo-HSCT—GVHD and infection. Despite variability across studies, treatment of patients with post-transplant complications utilized more HRU, with that use being greatest in patients experiencing severe or refractory forms of the complication.

Patients with aGVHD had higher HRU including longer initial and total LOS, higher rates of hospital readmission and ICU admissions, resulting in higher overall costs compared to those who did not experience aGVHD. Health resource requirements and costs were highest for patients experiencing more severe aGVHD (SR/HR). Similar patterns were evident with cGVHD, where the number and proportion of hospitalizations (including LOS per admission) were higher for patients experiencing cGVHD compared to non-cGVHD. Higher HRU requirements for more severe cGVHD resulted in higher costs.

CMV was associated with longer LOS, higher number, proportion, and duration of readmissions including a higher proportion and longer duration of ICU admissions compared to non-CMV. The greater need for HRU was reflected in higher total costs incurred by patients experiencing CMV complications compared to those who did not.

Our review found that the duration of letermovir therapy in CMV prophylaxis studies ranged from 69.4 to 100 days. Results from ongoing studies suggest that a doubling of that duration (to 200 days) may be of benefit to high-risk patients [66]. Given the treatment costs observed in studies reporting letermovir use (range USD 19,400 to USD 52,793), it is expected that doubling the duration of therapy would significantly impact associated costs.

While the costs and healthcare requirements are substantive for allo-HSCT, the incremental healthcare requirements and costs associated with the management of post-transplant complications are even greater [12,13]. Given the high number of patients experiencing fungal infections (9.2%) [67], CMVi (53.7%) [68], and aGVHD (31.4%), cGVHD (6.4%) or both (20.4%) [7], this has significant HRU and cost implications.

None of the previously published literature reviews identified through the search primarily focused on HRU and the costs associated with allo-HSCT complications. Therefore, it was not feasible to compare our findings to previous reviews. Similarly, synthesizing the results across studies to produce aggregate measures was not possible due to variations in the definition of the clinical outcome of interest (e.g., onset of cGVHD, classification of the CMV group and criteria for initiating Pre-ET), and how HRU and cost measures were presented. For example, whilst some studies presented the total HRU and cost for patients who experienced the complication compared to a control (usually patients without the complication), others presented HRU and costs directly associated with the complication of interest (without a control). Furthermore, while this review makes clear that allo-HSCT is associated with considerable HRU and costs, further studies will be needed to establish the impact of changes to allo-HSCT practice (such as increasing use of post-transplant cyclophosphamide) and the populations to whom it is offered (with increasing use in older age groups) [69].

There are several important limitations that impact the conclusions that can be drawn from this data. Firstly, given the diversity in the clinical context and measures of HRU and costs reported by the studies, definitive conclusions across the studies could not be drawn. Secondly, the methods used to derive HRU and cost results across the studies were not appraised. Assessment of the methodologies applied across studies was not feasible due to a lack of consistency in deriving these variables across studies. Thirdly, the search was carried out in November 2022; however, a subsequent updated review indicated that no significant research contradicting these findings has since been published. In addition, cost calculations could potentially be confounded by the choice of pricing index and the year in which it was applied. Finally, this review focuses on an assessment of HRU and costs and does not consider the clinical and patient-relevant outcomes produced associated with that care. It therefore does not provide a review of the value of care for complications of allo-HSCT.

## 5. Conclusions

This scoping review synthesizes the current literature on HRU and costs associated with post allo-HSCT complications. While the studies varied in their clinical context, reporting of HRU and cost measures, as well as the prevention and management strategies reported by the various centers, patients who experienced post allo-HSCT complications overall had higher HRU and incurred higher costs. Heterogeneity across studies prevented a more thorough synthesis of the direct HRU and costs associated with these complications. Developing a standardized framework for the conduct of HRU and cost analyses in allo-HSCT would better inform resource allocation in allo-HSCT, enabling clinicians, policymakers, and governments to design and deliver allo-HSCT care more efficiently.

## Figures and Tables

**Figure 1 curroncol-32-00283-f001:**
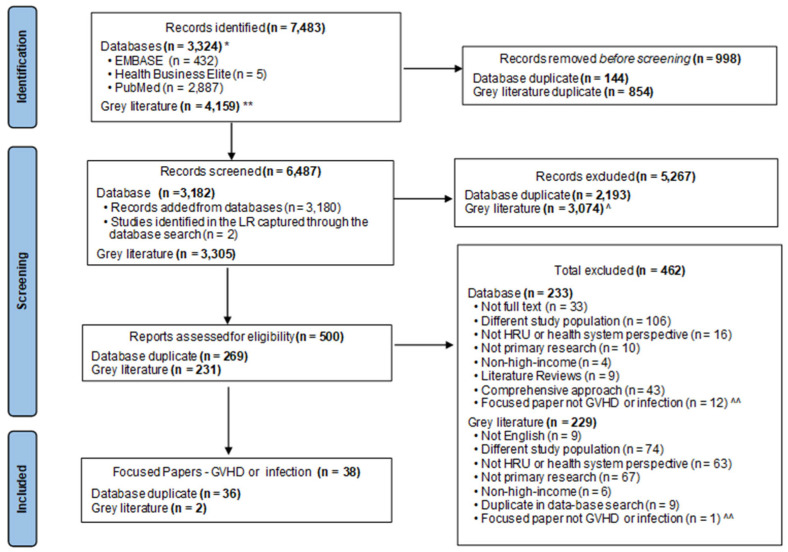
PRISMA (Exclusion rationale as per the Principal Reviewer). * search conducted 4 November 2022; ** search conducted 1 November 2022–2 November 2022; ^ pages without key terms ‘bone marrow transplant’, ‘blood and marrow transplant’, and ‘stem cell transplant’ and their abbreviations ‘BMT’ and ‘SCT’ in title; ^^ In Kim et al, 2024 [11] these papers formed part of the total excluded under ‘focused paper’. Abbreviations: GVHD = graft versus host disease; HRU = health resource utilization; LR = literature review.

**Figure 2 curroncol-32-00283-f002:**
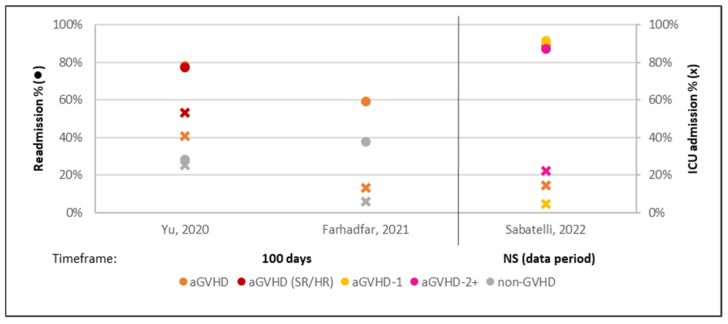
aGVHD readmission and ICU admissions. Key: ● = readmission rates, X = number of readmissions; references—Yu, 2020 [40], Farhadfar, 2021 [42], and Sabatelli, 2022 [41]. Abbreviations: aGVHD = acute graft versus host disease; aGVHD-1 = received one line of aGVHD treatment; aGVHD-2+ = received ≥two lines of aGVHD treatment; ICU = intensive care unit; NS = not specified; SR/HR = steroid-refractory or high-risk aGVHD.

**Figure 3 curroncol-32-00283-f003:**
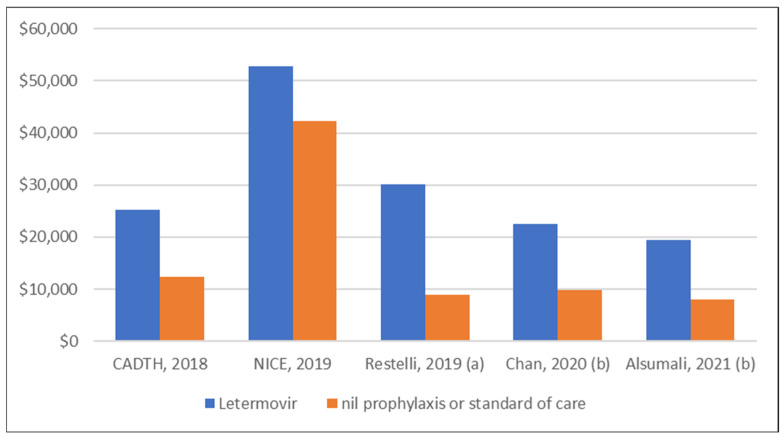
Studies presenting total costs per patient associated with letermovir prophylaxis. (a) costs were calculated based on the average of the two base case scenarios presented by the study, (b) per-patient costs were calculated based on study results which presented costs per 100 patients. References—CADTH, 2018 [47], NICE, 2019 [48], Restelli, 2019 [49], Chan, 2020 [50], and Alsumali, 2021 [51]. Abbreviations: CADTH = Canadian Agency for Drugs and Technologies in Health, NICE = National Institute of Health and Care Excellence, United Kingdom.

**Figure 4 curroncol-32-00283-f004:**
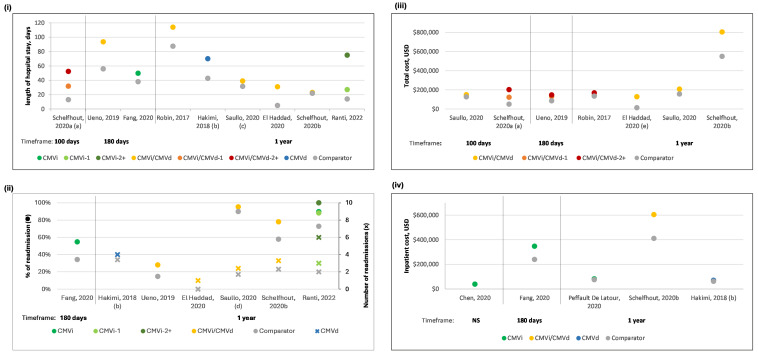
CMV hospitalization—length of stay (**i**), number and proportion of readmissions (**ii**), total cost (**iii**) and hospitalization cost (**iv**). Key: ● = readmission rates, X = number of readmissions. (a) study groups categorized by no, 1 and 2+ CMV readmissions; within 100 days of discharge from the index hospitalization (b) 1 year from first hospitalization with a diagnosis of CMVd (c) also presented transplant-related hospitalization (91.7 and 78.3 days, for csCMV and without csCMV), (d) proportion of any inpatient admissions and number of admissions (e) cost per encounter. References—(**i**) Schelfhout, 2020a [54], Ueno, 2019 [57], Fang, 2020 [52], Robin, 2017 [58], Hakimi, 2018 [48], Saullo, 2020 [59], El Haddad, 2020 [60], Schelfhout, 2020b [61], Ranti, 2022 [62] (**ii**) Fang, 2020 [52], Hakimi, 2018 [59], Ueno, 2019 [57], El Haddad, 2020 [60], Saullo, 2020 [56], Schelfhout, 2020b [61], Ranti, 2022 [62] (**iii**) Saullo, 2020 [56], Schelfhout, 2020a [54], Ueno, 2019 [57], Robin, 2017 [58], El Haddad, 2020 [60], Schelfhout, 2020b [61] (**iv**) Chen, 2020 [51], Fang, 2020 [55], Peffault De Latour, 2020 [56], Schelfhout, 2020b [61], and Hakimi, 2018 [59]. Abbreviations: CMV = Cytomegalovirus, CMVd = CMV disease, CMVi = CMV infection, NS = not specified, USD = United States dollars.

**Table 1 curroncol-32-00283-t001:** Summary of HRU and Cost measures reported in the included studies (≥ two studies).

Studies	Outcome	HRU/Cost	Complication/Intervention	Comparator	Implications
**Graft versus host disease**					
**aGVHD**					
Yu, 2019 [39]; Yu, 2020 [40]; Sabatelli, 2022 [41]; Farhadfar, 2021 [42]	LOS (initial)	HRU & Cost	↑aGVHD	Non-GVHD	↑↑ SR/HR ↓ for one line cf. ≥2 lines
	Total LOS	HRU & Cost	↑aGVHD	Non-GVHD	
	Readmission	HRU & Cost	↑aGVHD	Non-GVHD	HRU: % and length of readmission ↑↑ SR/HR
	ICU	HRU	↑aGVHD	Non-GVHD	HRU: % of patients
	Outpatient ^	Cost	↑aGVHD	Non-GVHD	
	Total	Cost	↑aGVHD	Non-GVHD	
	aGVHD-attributed	Cost	↑aGVHD	Non-GVHD	
**cGVHD**					
Scheid, 2022 [43]; Schain, 2021 [44]	Inpatient	HRU	↑cGVHD	Non-GVHD	HRU: % and no. of admissions ↑↑ SR/HR
	LOS	HRU	↑cGVHD	Non-GVHD	↑ based on cGVHD severity (severe, moderate & mild)
	Total	Cost	↑cGVHD	Non-GVHD	
**GVHD treatment**					
Yerrabothala, 2018 [45]; Boluda, 2021 [46]	LOS	HRU & Cost	↓ECP	Non-ECP	
	Inpatient	HRU	↓ECP	Non-ECP	HRU: % and no. of admissions
	Outpatient	HRU	↑ECP	Non-ECP	HRU: no. of visits
	Total	Cost	↓ECP	Non-ECP	
**Cytomegalovirus**					
**CMV Prophylaxis**					
CADTH, 2018 [47]; NICE, 2019 [48]; Restelli, 2019 [49]; Chan, 2020 [50]; Alsumali, 2021 [51]	Total	Cost	↑letermovir	SOC/placebo	
	Readmission	Cost	↓letermovir	SOC/placebo	
	Prophylaxis	Cost	↑letermovir	SOC/placebo	
	Pre-ET	Cost	↓letermovir	SOC/placebo	
	CMV Disease	Cost	↓letermovir	SOC/placebo	
**CMV infection or disease**					
Fang, 2020 [52]; Peffault De Latour, 2020 [53]; Schelfhout, 2020 [54]; Webb, 2018 [55]; Saullo, 2020 [56]; Ueno, 2019 [57]; Robin, 2017 [58]; Hakimi, 2018 [59]; El Haddad, 2020 [60]; Schelfhout, 2020 [61]; Ranti, 2022 [62];	LOS (initial)	HRU & Cost	↑CMVi/Pre-ET	No-CMV	↑ for one cf. ≥2 CMV-readmissions
	ICU	HRU	↑CMVi	No-CMV	HRU: length of admission ↓ length but ↑ proportion for one cf. ≥2 CMVi
	LOS	HRU & Cost	↑CMVi/CMVd	No-CMV ^^	↓ for one cf. ≥2 CMV-readmissions
	Readmission	HRU & Cost	↑CMVi/CMVd/Pre-ET	No-CMV	HRU: % and length of readmission ↓ for one cf. ≥2 CMV-readmissions
	Outpatient	HRU & Cost	↑CMVi/CMVd	No-CMV	HRU: % and no. of visits ↓ for one cf. ≥2 CMVi/CMVd
	Total	Cost	↑CMVi/CMVd	No-CMV	↓ for one cf. ≥2 CMVi/CMVd
	Antiviral therapy	HRU & Cost	↑CMVi/CMVd	No-CMV	HRU: % Pre-ET antiviral therapy ↓ for one cf. ≥2 CMVi

Key: ↑/↓ increase/decreased HRU and/or cost with complication/intervention compared to comparator. ^ refers to first 100 days, outpatient cost slightly lower for non-aGVHD compared to aGVHD at Day 101–365; ^^ control group for one study was GVHD. Abbreviations: aGVHD = acute graft versus host disease; CADTH = Canadian Agency for Drugs and Technologies in Health; cf = compared; cGVHD = chronic graft versus host disease; CMV = Cytomegalovirus; CMVd = Cytomegalovirus disease; CMVi = Cytomegalovirus infection; ECP = Extracorporeal Photopheresis; GVHD = graft versus host disease; HRU = health resource utilization; ICU = intensive care unit; LOS = length of stay; NICE = National Institute for Health and Care Excellence, United Kingdom; Pre-ET = pre-emptive therapy; SR/HR = steroid-refractory or high-risk; SOC = standard of care.

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
