# Peer review of "Healthcare Resource Use and Costs of Allogeneic Hematopoietic Stem Cell Transplantation Complications: A Scoping Review"

_curroncol, 2025, doi:10.3390/curroncol32050283_

Round 1
Reviewer 1 Report
Comments and Suggestions for Authors
The manuscript by Kim et al., titled "Healthcare resource use and costs of allogeneic hematopoietic stem cell transplantation complications: a scoping review," provides a literature review on healthcare resource utilization and costs associated with complications from allogeneic hematopoietic stem cell transplantation (allo-HSCT). The manuscript aims to synthesize available evidence on the economic burden and healthcare demand related to these complications, particularly infections and graft-versus-host disease (GvHD), offering a comprehensive overview that may support resource allocation and health policy decision-making.
Points that require improvement in the manuscript:
- Regarding the introduction, it could better contextualize the impact of allo-HSCT complications on the global healthcare system. Additionally, some paragraphs are lengthy and could be reorganized for enhanced clarity, providing the reader with a more engaging narrative.
- Although the criteria are described, the lack of detailed explanations regarding their definitions could compromise the reproducibility of the study. Clearly describing the rationale behind the chosen parameters is essential.
- The conversion of costs to 2022 USD is appropriate; however, the standardization methodology is not fully detailed. It is necessary to improve and clarify this methodology.
- The manuscript could further explore direct comparisons between the current findings and previous studies, highlighting similarities and differences in cost estimates and healthcare resource utilization.
- While addressing the economic impact of allo-HSCT complications, the manuscript could discuss in greater detail how these findings might influence reimbursement policies and hospital resource planning.
- Limitations are mentioned but not discussed thoroughly. It would be helpful to include a critical analysis of potential biases in study selection and how these might have affected the results.
- It would be beneficial to generate graphs or figures from the tables to facilitate better understanding of the presented information.
- The conclusion could be enhanced by clearly highlighting the main implications of the findings for clinical practice and healthcare policies, as well as suggesting directions for future research in this field.
Reviewer 2 Report
Comments and Suggestions for Authors
In the submitted manuscript, the authors present a comprehensive review of healthcare resource utilization (HRU) and costs associated with complications of allogeneic hematopoietic stem cell transplantation (allo-HSCT), with a focus on graft-versus-host disease (GVHD) and infections. The authors mainly focus are two questions: What are the HRU and costs related to GVHD and infections after allo-HSCT? How do such complications impact healthcare costs and resource use compared to uncomplicated transplants?
Overall, the quality of the manuscript is good, with the data presented clearly. The search strategy for studies and literatures are detailed and comprehensive. Standardization of reported costs to 2022 USD for comparability is also well-noted.
A few points should be considered to strengthen the paper:
- The authors should provide more background introduction on the current state of allo-HSCT and how elaborating HRU and costs could impact decision-making.
- The authors should explain in more detail how data from different studies were combined and analyzed.
